# A Study of the MTHFR Gene Prevalence in a Rural Tennessee Opioid Use Disorder Treatment Center Population

**DOI:** 10.3390/ijerph19063255

**Published:** 2022-03-10

**Authors:** Leslie Cole, Alina Cernasev, Katie Webb, Santosh Kumar, A. Shaun Rowe

**Affiliations:** 1Pathway Healthcare, LLC, 801 Hill Street, Springfield, TN 37172, USA; lesliesccole@icloud.com; 2Department of Clinical Pharmacy and Translational Science, College of Pharmacy, University of Tennessee Health Science Center, 301 S. Perimeter Park Drive, Suite 220, Nashville, TN 37211, USA; 3Pathway Healthcare, LLC, 1000 Urban Center Drive, Suite 600, Birmingham, AL 35242, USA; kwebb@pathwayhealthcare.com; 4Department of Pharmaceutical Sciences, College of Pharmacy, University of Tennessee Health Science Center, 881 Madison Ave, Memphis, TN 38163, USA; ksantosh@uthsc.edu; 5Department of Clinical Pharmacy and Translational Science, College of Pharmacy, University of Tennessee Health Science Center, 1924 Alcoa Hwy, Box 117, Knoxville, TN 37920, USA; arowe@uthsc.edu

**Keywords:** opioid use disorder, Methylenetetrahydrofolate reductase (MTHFR) gene, prevalence, addiction

## Abstract

*Background:* Opioid Use Disorder (OUD) has been linked to dopamine and the neurological reward centers. Methylenetetrahydrofolate reductase (MTHFR) is an enzyme involved in the production of many neurotransmitters such as dopamine. As such, MTHFR variants that lead to decreased production of neurotransmitters may play a role in OUD. However, lacunae exist for characterizing the prevalence of the MTHFR mutations in an OUD population. The objective of this study was to determine prevalence of the MTHFR gene mutations in a rural Tennessean population with OUD. *Methods:* This study was a retrospective cohort of individuals with OUD that evaluated the prevalence of MTHFR variants. Patients were categorized as normal, homozygous C677T, heterozygous C677T, homozygous A1298C, or heterozygous A1298C. The primary outcome was a qualitative comparison of the prevalence of each of the MTHFR variants in our cohort to the publicly reported MTHR polymorphism prevalence. Secondary outcomes include race and ethnicity differences as well as stimulant use differences for each of the variants. *Results:* A total of 232 patients undergoing care for opioid use disorder were included in the study. Of those included, 30 patients had a normal MTHFR allele and 202 had a variant MTHFR allele. Overall, the prevalence of any MTHFR variant was 87.1% (95% CI 82.6–91.4%). When comparing those with a normal MTHFR allele to those with any MTHFR variant, there was no difference in age, sex, race and ethnicity, or stimulant use. *Conclusion:* The overall prevalence of MTHFR variants in patients with opioid use disorders is high.

## 1. Introduction

The opioid epidemic rages on despite efforts to mitigate it. Misuse of opioids, including pain medication, heroin, and synthetic opioids, has been linked as a primary risk factor for the development of opioid use disorder (OUD) [1,2,3]. Opioids, both synthetic and non-synthetic, remain the leading cause of drug overdose deaths [4]. According to the CDC, the reported number of drug overdose deaths in the United States in a 12-month period has risen from 72,124 deaths for the 12 months preceding January of 2020 to 94,134 deaths for the 12 months preceding January 2021. This represents a 30.5% increase in reported overdose deaths in the United States from January 2020 to January 2021. In Tennessee, there has been a 48.2% increase in reported drug overdose deaths during the same time frame [5]. As evidenced by these statistics, the burden of OUD is continuing to expand. As such, research to understand the underlying mechanisms of disease in OUD is needed.

OUD has clearly been linked to dopamine and the neurological reward centers [6]. There has been evidence suggesting a link between OUD, dopamine receptors, and the genes that encode the dopamine receptors [7,8,9,10]. The standard outpatient treatment of OUD incorporates the use of medication with behavioral counseling [11]. Buprenorphine, a medication that binds to opioid receptors, has been shown to decrease the risk of relapse on opioids, including heroin [12]. However, despite the use of buprenorphine, many individuals continue to look for addictive substances, such as cocaine and methamphetamine [13]. This suggests a potential absence of neurological reward despite adequate buprenorphine treatment.

Methylenetetrahydrofolate reductase (MTHFR) is an enzyme that converts 5,10-methylenetetrahydrofolate to 5-methyltetrahydrofolate [14]. MTHFR’s methylation of folate is a key step in the overall process of methylation. It has been demonstrated that a polymorphism of the gene encoding MTHFR at either the C677T or A1298C site can cause a deficiency in the MTHFR enzymatic activity [15]. MTHFR enzymatic activity is vital to the optimal production of several key neurotransmitters including serotonin, dopamine, glutamate, and GABA [16,17]. These neurotransmitters play an important role in the biological responses in addiction [6].

Previous studies have demonstrated a link between the presence of MTHFR gene polymorphisms and several psychiatric conditions, including schizophrenia, bipolar disorder, depression, and attention deficit disorder [14,18,19]. Of note, these psychiatric conditions are common comorbidities in patients with opioid use disorder [18].

In the same vein, an additional association has also been suggested between the treatment of pain and the prevalence of MTHFR genetic polymorphisms [20]. DNA methylation is associated with MTHFR enzymatic activity [21]. Previous research has also connected chronic pain and OUD [22,23]. However, there remain lacunae in patient data regarding the prevalence of the MTHFR gene mutation and OUD, which presents challenges in providing adequate health services to patients in the United States. One study conducted in North Carolina demonstrated a higher prevalence of the MTHFR gene variants in patients with OUD [24]. Another study established a higher prevalence of MTHFR C677T gene mutations in individuals who use heroin [24]. However, a paucity of data exists for further characterizing the prevalence of MTHFR gene mutation and its implications for practicing clinicians. Thus, this study’s objectives are to determine the prevalence of the MTHFR gene mutations, C677T and A1298C, in an OUD population in rural Tennessee, and explore their roles on OUD.

## 2. Materials and Methods

This was an Institutional Review Board (IRB) approved exempt retrospective cohort study (IRB# 21-08253-XP). Consecutive patients from July 2018 to July 2021 who received treatment for OUD at an outpatient behavioral health center focusing on addiction treatment in a rural Tennessee community were considered for inclusion in the analysis. Patients with OUD who were 18 years old or greater and had genetic testing for the MTHFR variants C677T and A1298C were included. Patients were tested for these gene variants as a part of their routine care based upon clinical symptoms (e.g., low energy, low motivation) and the known prevalence of comorbid conditions associated with MTHFR mutations (e.g., depression, bipolar disorder) with OUD. The intention of these measurements was to supplement with methylated folate or SAM-e if necessary. Those patients without genetic testing for MTHFR variants were excluded from the study. As the information for this study was collected during the routine care of patients and was deidentified before inclusion in the study, the IRB granted a waiver of informed consent.

Baseline demographics on the cohort included age, sex, race and ethnicity, and stimulant use. Stimulant use was defined as documentation in the medical record from self-reporting or a urinalysis positive for stimulants. In addition, the results of genetic testing for the MTHFR variants (C677T and A1298C) were abstracted from the medical record. Patients were categorized into the following groups: normal (no variant), homozygous C677T (homozygous for C677T with a normal A1298C), heterozygous C677T (Heterozygous for C677T and either normal or heterozygous A1298C), homozygous A1298C (homozygous for A1298C with a normal C677T), or heterozygous A1298C (Heterozygous for A1298C and either normal or heterozygous C677T).

The primary outcome of this study was a qualitative comparison of the prevalence of each of the MTHFR variants in our cohort to the publicly reported MTHFR polymorphism prevalence [25]. Secondary outcomes include race and ethnicity differences as well as stimulant use differences for each of the variants.

All statistical analysis was conducted with SAS (SAS Institute Inc., Cary, NC, USA; Version 9.4 [TS1M7]). Continuous variables were checked for normality with a Shapiro–Wilk Test, Kolmogorov–Smirnov Test, visual evaluation of histograms, and QQ plots. Those variables found to have a normal distribution are presented as mean and standard deviation. Between group comparisons for normally distributed variables were made with a Student’s *t*-test. Categorical variables are presented as count and proportion of group. Between group comparisons are made with a chi-square test or Fisher’s Exact as appropriate based on expected cell counts. For between group comparisons, the cohort was dichotomized based on MTHFR variant status. The comparison group for all between group comparisons were those patients with the normal MTHFR allele. Statistical significance was defined as a *p* value less than 0.05. Prevalence was calculated as the ratio of patients with MTHFR variants to the total population studied. A 95% confidence interval was calculated for each prevalence using the Wald method without continuity correction.

## 3. Results

A total of 232 patients undergoing care for opioid use disorder were included in the study. Of those included, 30 patients had a normal MTHFR allele and 202 had a variant MTHFR allele. Overall, the prevalence of any MTHFR variant was 87.1% (95% CI 82.6–91.4%). When comparing those with a normal MTHFR allele to those with any MTHFR variant, there was no difference in age, sex, race and ethnicity, or stimulant use. Complete comparisons of the cohort can be seen in Table 1.

### 3.1. Homozygous C677T

The prevalence of having a homozygous c677t mutation is 9.5% (95% CI: 5.7–13.3%). This is similar to the overall reported population prevalence of 10.9% (9.4% to 12.5%). Overall, there was no difference in age, sex, race, or stimulant use when comparing those with a homozygous C677T variant to those with a normal variant. A complete comparison of homozygous C677T to those with a normal MTHFR allele can be seen in Table 2.

### 3.2. Heterozygous C677T

The prevalence of having a heterozygous C677T mutation is 49.1% (95% CI: 42.7–55.6%). This is higher than the overall reported population prevalence of 39.8% (37.2–42.5%). There were no differences in age, sex, race, or stimulant use when comparing tot those patients with a normal variant. A complete comparison of heterozygous C677T to those with a normal MTHFR allele can be seen in Table 3.

### 3.3. Homozygous A1298C

The prevalence of having a homozygous a1298c mutation is 6.5% (95% CI: 3.3–9.6%). This is lower than the overall reported population prevalence of 9% (7.4–10.9%). There was no difference in age, sex, or race. However, as compared to those patients with a normal allele, a significantly lower proportion of patients in the homozygous a1298c allele group had stimulant use (14 [46.7%] vs. 2 [13.3%]; *p* = 0.0277). A complete comparison of homozygous A1298C to those with a normal MTHFR allele can be seen in Table 4.

### 3.4. Heterozygous A1298C

The prevalence of having a heterozygous A1298C mutation is 47.4% (95%CI: 41.0–53.8%). This is higher than the overall reported population prevalence of 38.8% (36.1–41.5%). Overall, there was no difference in age, sex, race, or stimulant use when comparing those with a heterozygous A1298C variant to those with a normal variant. A complete comparison of heterozygous A1298C to those with a normal MTHFR allele can be seen in Table 5.

## 4. Discussion

This study examined the prevalence of MTHFR mutations in individuals with opioid use disorders and whether these mutations may be associated with any clinical implications. Our data from individuals with OUD on the prevalence of specific variants show both similarities and differences with reported prevalence of these specific mutations in general population [25]. In general populations, 60–70% of individuals have at least one of these variants, 8.5% have *Homozygous C677T or A1298C*, and 2.25% have *Heterozygous C677T*
*or A1298C* [26].

The vital role of the MTHFR enzyme in the methylation of folate raises the theoretical opportunity for clinical intervention. For example, previous studies have described the link between MTHFR polymorphisms, folate serum levels, and the response to methotrexate (MTX) treatment for certain types of cancer [27,28]. The implication that relationships exist between medications, folate levels, and MTHFR polymorphisms creates an opportunity for future studies to evaluate such relationships in patients undergoing treatment for OUD. To date, there are no data to confirm a connection between MTHFR and opioid medications. One case report of a patient with MTHFR polymorphism demonstrated a substantial reduction in pain severity following supplementation with folinic acid [29]. The study recommended that physicians initiate pharmacogenomic testing for MTHFR polymorphism, especially for those patients who are not responding to standard medications [29]. In the same vein, a study evaluated the link between disulfiram treatment that is used for the treatment of cocaine addiction and MTHFR polymorphism [30]. Results from a study by Spellicy et al. suggested that pharmacogenomic testing for the MTHFR C677T polymorphism represents a valuable resource to detect patients who might demonstrate an enhanced response to disulfiram treatment for cocaine addiction [30]. There are minimal data to show a relationship between MTHFR gene polymorphism and other substances of abuse. Of note, one study examined the link between MTHFR C677T single nucleotide polymorphism and hepatocellular carcinoma in participants who had alcoholic liver disease and received a liver transplant [31]. This study showed an enhanced risk of developing hepatocellular carcinoma in men diagnosed with alcoholic cirrhosis [31]. This further demonstrates the potential clinical implications of these polymorphisms on disease severity and risk of complications for individuals with substance use disorders. As a result, our study supports a greater probability of a connection between OUD and the MTHFR polymorphisms. It would be valuable to pursue further research evaluating the possible impacts of MTHFR polymorphisms on individuals with OUD.

### 4.1. Strength and Limitations

One of the strengths of this study is that it is the first study of its kind conducted in Tennessee, a state that has seen a significant increase in reported drug overdose deaths in recent years [5]. However, some caveats must be noted when interpreting the present findings. Notably, the convenience samples used in this study may limit the generalization of the results to the broader OUD population. More specifically, due to the lack of sufficient subject populations of different ethnic groups and genders, we cannot delineate the effect of MTHFR variants on OUD by ethnic groups or genders. The study also could not delineate the effect of MTHFR polymorphisms on OUD in the absence or presence of other substances of abuse, which are commonly prevalent with OUD populations [32]. This study specifically evaluated a convenient population of individuals with OUD because this is the population served by the investigators. The individuals in this sample population were tested for MTHFR polymorphism as a part of their routine care, based upon observed clinical symptoms and the known prevalence of comorbid conditions associated with MTHFR mutations in individuals with OUD. The intention of these measurements was to supplement with methylated folate or SAM-e, if necessary. Ultimately, this study demonstrates the need to design a more robust study enrolling a large cohort of individuals with sufficient numbers of different ethnic groups and genders from various geographical locations to generalize the conclusions reached in this study. Further, additional research with a larger cohort of subjects is necessary to better understand the potential clinical implications of MTHFR polymorphisms in individuals with OUD and other substance use disorders.

### 4.2. Clinical Implications

Our study has clinical relevance because the MTHFR C677T polymorphism has a relatively higher prevalence in opioid users than the general population. The implications include the possibility that the MTHFR C677T polymorphism could pose a risk for development of OUD, in which case, testing for the mutation could serve as a predictor or even a preventative screening tool for individuals being prescribed opioids. Furthermore, the results of this study also have clinical implications for future treatment options. For example, the results highlight the need for an evaluation of whether methylated folate may be an adjunctive treatment in individuals with OUD who have the MTHFR C677T polymorphism.

## 5. Conclusions

The study reveals that individuals with OUD have a relatively high prevalence of MTHFR variants. While we were unable to find an association between MTHFR polymorphisms and clinical outcomes in individuals with OUD, these findings will contribute to future research.

## Figures and Tables

**Table 1 ijerph-19-03255-t001:** Overall comparison of normal MTHFR allele and any variant MTHFR allele.

Variable	MTHFR Normal *n* = 30	MTHFR Abnormal *n* = 202	*p*-Value
Age, years, mean (SD) ^†^	38.8 (10.2)	38.6 (10.4)	0.9515
Male, *n* (%) ^‡^	9 (30)	41 (20.3)	0.2278
Race and Ethnicity, *n* (%)			>0.05
White *	26 (86.7)	180 (89.1)	
Black *	1 (3.3)	3 (1.5)	
American Indian *	0 (0)	1 (0.5)	
Unknown Race *	3 (10)	17 (8.4)	
Other Race *	0 (0)	1 (0.5)	
Hispanic *	1 (3.3)	6 (3.0)	
Stimulant Use ^‡^	14 (46.7)	71 (35.2)	0.2218

^†^ Student’s *t*-test, ^‡^ Chi Square, * Fisher’s Exact.

**Table 2 ijerph-19-03255-t002:** Homozygous C677T compared to normal MTHFR allele.

Variable	Normal *n* = 30	Homozygous c677t *n* = 22	*p*-Value
Age, years, mean (SD) ^†^	38.8 (10.2)	39.5 (10.4)	0.7517
Male, *n* (%) ^‡^	9 (30)	5 (22.7)	0.5591
Race and Ethnicity, *n* (%)			>0.05
White *	26 (86.7)	19 (86.4)	
Black *	1 (3.3)	0 (0)	
American Indian	0 (0)	0 (0)	
Unknown race *	3 (10)	3 (13.6)	
Other race	0 (0)	0 (0)	
Hispanic *	1 (3.3)	0 (0)	
Stimulant Use ^‡^	14 (46.7)	12 (54.2)	0.5745

^†^ Student’s *t*-test, ^‡^ Chi Square, * Fisher’s Exact.

**Table 3 ijerph-19-03255-t003:** Heterozygous C677T compared to those with a normal MTHFR allele.

Variable	Normal *n* = 30	Heterozygous c677t *n* = 114	*p*-Value
Age, years, mean (SD) ^†^	38.8 (10.2)	39.2 (10.2)	0.8357
Male, *n* (%) ^‡^	9 (30)	26 (22.8)	0.4138
Race and Ethnicity, *n* (%)			>0.05
White *	26 (86.7)	103 (90.4)	
Black *	1 (3.3)	1 (0.9)	
American Indian *	0 (0)	1 (0.9)	
Unknown race *	3 (10)	8 (7.0)	
Other race *	0 (0)	1 (0.9)	
Stimulant use ^‡^	14 (46.7)	37 (32.5)	0.1476

^†^ Student’s *t*-test, ^‡^ Chi Square, * Fisher’s Exact.

**Table 4 ijerph-19-03255-t004:** Homozygous A1298C compared to those with a normal MTHFR allele.

Variable	Normal *n* = 30	Homozygous a1298c *n* = 15	*p*-Value
Age, years, mean (SD) ^†^	38.8 (10.2)	38.5 (12.7)	0.7517
Male, *n* (%) ^‡^	9 (30)	1 (6.7)	0.1288
Race and Ethnicity, *n* (%)			> 0.05
White *	26 (86.7)	13 (86.7)	
Black *	1 (3.3)	1 (6.7)	
American Indian	0 (0)	0 (0)	
Unknown race *	3 (10)	1 (6.7)	
Other race	0 (0)	0 (0)	
Hispanic *	1 (3.3)	0 (0)	
Stimulant Use ^‡^	14 (46.7)	2 (13.33)	0.0277

^†^ Student’s *t*-test, ^‡^ Chi Square, * Fisher’s Exact.

**Table 5 ijerph-19-03255-t005:** Heterozygous A1298C compared to those with a normal MTHFR allele.

Variable	Normal *n* = 30	Heterozygous A a1298c *n* = 110	*p*-Value
Age, years, mean (SD) ^†^	38.8 (10.2)	39.1 (10.2)	0.8841
Male, *n* (%) ^‡^	9 (30)	21 (19.1)	0.1968
Race and Ethnicity, *n* (%)			>0.05
White ^‡^	26 (86.7)	101 (91.8)	
Black *	1 (3.3)	1 (0.9)	
American Indian *	0 (0)	1 (0.9)	
Unknown race *	3 (10)	7 (6.4)	
Other race	0 (0)	0 (0)	
Hispanic *	1 (3.3)	6 (5.5)	
Stimulant Use ^‡^	14 (46.7)	43 (39.1)	0.4541

^†^ Student’s *t*-test, ^‡^ Chi Square, * Fisher’s Exact.

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
