# Peer review of "A Study of the MTHFR Gene Prevalence in a Rural Tennessee Opioid Use Disorder Treatment Center Population"

_ijerph, 2022, doi:10.3390/ijerph19063255_

Round 1
Reviewer 1 Report
This is a study which aims to estimate the allele frequencies at certain MTHFR gene variants, within people with OUD from Tennessee. The paper is well written and concise, however here are some suggestions for improvement:
- As the study is looking at gene prevalence in a population with OUD specifically, this should be made clear in the title. The more general 'MTHR gene prevalence in rural Tennessee community' is too general and is somewhat misleading.
- Why is the study being termed a 'pilot study' ? This usually suggests a study that will be used to inform a future, larger study. Is this the case for the current study ?
- Lines 81-82: please revise the wording '...who received care who received treatment...' Should there be an 'and' here between 'care' and 'who' ?
- Can you be more specific than "health center in the Southeastern United States" when describing the study's location ?
- Were patients genotyped for the two variants of interest as part of their routine care ? If so, please clarify why this was the case.
- It is not clear what you mean by "we collected the results of
genetic testing for the MTHFR variants study evaluating the prevalence of MTHFR variants C677T and A1298C." in lines 91-92, since the whole study is for evaluating the prevalences. The sentence doesn't appear to make sense. - Please ensure that you justify the sample size in terms of precision of estimates of prevalence and also power for the analyses of association.
- For the Fisher's exact test/chi-square test for difference in ethnicity/race only one p-value should be reported, as this is a categorical variable with several categories, not a series of many binary variables.
- Please specify what your threshold for statistical significance is, accounting for the multiple tests conducted.
- Testing separately for each genotype in turn vs 'normal' (Tables 2-5) is not an efficient way of testing for association. It introduces multiple testing problems and also does not make sense as all possible genotypes are categories of one categorical variable "genotype". It would be better to conduct one Chi-squared test (or anova for continuous variable Age) for each variable e.g. for sex you would have sex vs genotype (with genotype having all possible genotypes as categories) and test using the chi-squared test. You would then only have two tables - one comparing normal to variant carriers and one comparing across all genotype categories.
- What ethnic group have the population prevalences you are comparing to been calculated in ? It is important that this is consistent with the underlying ethnicities of the study population. You might wish to restrict you analysis to whites only, or undertake separate comparisons again population prevalence for each ethnic group separately (where there are sufficient numbers). As allele frequencies are known to vary across ethnic groups it is important to compare like with like.
- In line 160 you state "and its clinical implications for patients taking pain medications". I cannot see how your study assessed this aspect. Can you please clarify ?
- As you focus quite significantly in the Discussion on the fact that prevalence of C677T heterozygotes is higher than the general population, it would have been useful to satistically compare the various prevalences to the population prevalences to assess whether the differences were statistically significant.
- Your discussion focusses a lot on the previous literature between the variants investigated and various clinical outcomes. This seems to be disproportionate to the simplicity of your study and the fact that there was no intention of trying to associate genotype in the current study with any clinical outcomes. It would be useful to mention a few implications of having a particular genotype on risk of certain clinical outcome, but I do not feel it is appropriate to discuss this in length since it is out of the remit of the current study. To this end, I would recommend simplifying the Discussion and Conclusion sections significantly to better reflect the context of the current study and it's aims.
Author Response
appreciate your concern and we have removed the word, “pilot study”.
- Lines 81-82: please revise the wording ‘...who received care who received treatment...’ Should there be an ‘and’ here between ‘care’ and ‘who’ ?
Response: Thank you for bringing this to our attention. The typo has been corrected.
- Can you be more specific than “health center in the Southeastern United States” when describing the study’s location ?
Response: Thank you for this suggestion. Line 87-88 has been amended to more specifically describe the setting: “…outpatient behavioral health center focusing on addiction treatment in a rural Tennessee community”.
- Were patients genotyped for the two variants of interest as part of their routine care ? If so, please clarify why this was the case.
Response: Thank you for this question. Patients were tested for these gene variants as a part of their routine care based upon clinical symptoms (e.g., low energy, low motivation) and the known prevalence of comorbid conditions associated with MTHFR mutations (e.g., depression, bipolar disorder) with OUD. The intention of these measurements was to supplement with methylated folate or SAM-e if necessary. Specific variants (C677T & A1298C) were measured based upon laboratory limitations. - It is not clear what you mean by “we collected the results of genetic testing for the MTHFR variants study evaluating the prevalence of MTHFR variants C677T and A1298C.” in lines 91-92, since the whole study is for evaluating the prevalences. The sentence doesn’t appear to make sense.
Response: The authors would like to thank you for your recommendation. We have modified the sentence for clarity. It now reads, “In addition, the results of genetic testing for the MTHFR variants (C677T and A1298C) were abstracted from the medical record.”
- Please ensure that you justify the sample size in terms of precision of estimates of prevalence and also power for the analyses of association.
Response: The authors would like to thank you for this recommendation. As this was a convenience sample, there was not an a priori power analysis conducted. We have updated the methods to indicate that 95% confidence intervals were calculated for each prevalence.
- For the Fisher’s exact test/chi-square test for difference in ethnicity/race only one p-value should be reported, as this is a categorical variable with several categories, not a series of many binary variables.
Response: The authors would like to thank the reviewer for his or her observation. This has been updated throughout the manuscript. This observation has strengthened our manuscript. Thank you.
- Please specify what your threshold for statistical significance is, accounting for the multiple tests conducted.
Response: Thank you for pointing out this omission. We have updated the methods to include the specified alpha value.
- Testing separately for each genotype in turn vs ‘normal’ (Tables 2-5) is not an efficient way of testing for association. It introduces multiple testing problems and also does not make sense as all possible genotypes are categories of one categorical variable “genotype”. It would be better to conduct one Chi-squared test (or anova for continuous variable Age) for each variable e.g. for sex you would have sex vs genotype (with genotype having all possible genotypes as categories) and test using the chi-squared test. You would then only have two tables – one comparing normal to variant carriers and one comparing across all genotype categories.
Response: The authors would like to thank the reviewer for his or her comments. While we respect your opinion on this matter, the overall effect of modifying our statistical analysis plan is moot. As we did not find a statistical difference for the vast majority of our groupings, the use of an ANOVA or Chi-square without the pairwise comparisons would not have found a difference either. Our goal was to conduct a pair-wise comparison of all genotypes tested in order to provide an overall view of the groupings.
- What ethnic group have the population prevalences you are comparing to been calculated in ? It is important that this is consistent with the underlying ethnicities of the study population. You might wish to restrict you analysis to whites only, or undertake separate comparisons again population prevalence for each ethnic group separately (where there are sufficient numbers). As allele frequencies are known to vary across ethnic groups it is important to compare like with like.
Response: We value your thoughtful suggestions. Our population size is much smaller and thus it might not be reasonable to compare the to the ethnicity-specific data from the CDC. The sample size is not sufficient to do an analysis for each ethnic group. Therefore, it is a valuable suggestion to do a separate analysis based on “whites” only and write a sentence, if the results are similar. Perhaps, there is no significant difference when we analyze the entire group vs. whites only. Thus, in our follow up study, we will increase the study population with sufficient number of subjects from each ethnic group and analyze them separately.
- In line 160 you state “and its clinical implications for patients taking pain medications”. I cannot see how your study assessed this aspect. Can you please clarify ?
Response: Thank you for this comment. This sentence was removed and the previous sentence was edited to say “This study examined the prevalence of MTHFR mutations in individuals with opioid use disorders and whether these mutations may be associated with any clinical implications.”
- As you focus quite significantly in the Discussion on the fact that prevalence of C677T heterozygotes is higher than the general population, it would have been useful to satistically compare the various prevalences to the population prevalences to assess whether the differences were statistically significant.
Response: The authors would like to thank the reviewer for his or her comments. We agree with this assessment, however, the publicly available data only listed prevalence in percentage without providing numbers of patients tested. As such were unable to do more than conduct a descriptive comparison.
- Your discussion focusses a lot on the previous literature between the variants investigated and various clinical outcomes. This seems to be disproportionate to the simplicity of your study and the fact that there was no intention of trying to associate genotype in the current study with any clinical outcomes. It would be useful to mention a few implications of having a particular genotype on risk of certain clinical outcome, but I do not feel it is appropriate to discuss this in length since it is out of the remit of the current study. To this end, I would recommend simplifying the Discussion and Conclusion sections significantly to better reflect the context of the current study and it’s aims.
Response: We value your suggestions and we agree with the reviewer. Accordingly, we have significantly reduced the discussion section. For examples, we have removed the last past of the first paragraph (lines 177-184 and lines 186-187). We have also completely removed paragraph 2 of the discussion (lines 188-199). The fourth paragraph of discussion (lines 213-225) was significantly edited to fit the discussion just on alcohol.
Reviewer 2 Report
Overall it is an interesting paper discussing the association between MTHFR variant and OUD. However in Materials and Methods, there seems to be overlaps in patients categorization: patients that are heterozygous for C677T and heterozygous for A1298C are categorized into two groups, Group 'Heterozygous C677T' and Group 'Heterozygous A1298C'. Besides, since there's a significant lower proportion of patients in the homozygous a1298c allele group for stimulant use, it would be great if the authors discuss more on this.
Author Response
Overall it is an interesting paper discussing the association between MTHFR variant and OUD. However in Materials and Methods, there seems to be overlaps in patients categorization: patients that are heterozygous for C677T and heterozygous for A1298C are categorized into two groups, Group ‘Heterozygous C677T’ and Group ‘Heterozygous A1298C’. Besides, since there’s a significant lower proportion of patients in the homozygous a1298c allele group for stimulant use, it would be great if the authors discuss more on this.
Response: Thank you for this valuable suggestion. We amended the text and your suggestions strengthen our manuscript.
Reviewer 3 Report
Manuscript review for IJERPH
Comments to authors
Manuscript Number: IJERPH- 1574275
“MTHFR gene prevalence in rural Tennessee community: A pilot study”
February 10, 2022
This study aims to assess and compare prevalence of MTHFR gene polymorphism in rural Tennesseans with opioid use disorder (OUD). The study provides some interesting descriptive data but the manuscript is poorly focused and lacks clarity regarding the study objective, hypotheses, and motivation. Further, the statistical analysis is rudimentary and fails to shed any light on the potential relationship between MTHFR genotypes and OUD, while discussion of findings is meandering and untethered to any specific study objective that is laid out in advance. These issues are discussed below in more detail:
- Please provide a clear study objective and a crisply articulated rationale for studying the comparative prevalence of MTHFR polymorphisms in your study population. It seems the study could be presented simply as a brief paper focused on providing raw prevalence of these polymorphisms in Tennessee (as the title suggests). If that is the case, authors need to explain why they chose to assess people with OUD and not the general population for this screening study.
- Explain the limitations of your convenience sample more clearly in the limitations section. It seems only people with OUD who agreed to be tested for polymorphism were include in the study, raising selection issues that affect whether the sample is representative of the target population.
- Are you able to say anything regarding the relationship between OUD severity, profile and prevalence of specific types of MTHFR polymorphism? If not, please explain why you do not include data on clinical profile of OUD patients for the clinic when you have access to medical records.
- The discussion of clinical implications of your findings for conditions unrelated to OUD e.g., autism, cancer is misplaced and should be removed. At the very least the connection should be anticipated in study objectives and clearly tied to your findings.

Author Response
- comparative prevalence of MTHFR polymorphisms in your study population. It seems the study could be presented simply as a brief paper focused on providing raw prevalence of these polymorphisms in Tennessee (as the title suggests). If that is the case, authors need to explain why they chose to assess people with OUD and not the general population for this screening study.
Response: Thank you for this comment and suggestion. The study objective in the abstract has been updated to more clearly reflect the objective of this study. This study specifically evaluated a population of individuals with OUD because that is the population served by the primary investigator. The provider began testing for this genetic abnormality based upon clinical symptoms (e.g., low energy, low motivation) and the known prevalence of comorbid conditions associated with MTHFR mutations (e.g., depression, bipolar disorder) with OUD.
- Explain the limitations of your convenience sample more clearly in the limitations section. It seems only people with OUD who agreed to be tested for polymorphism were include in the study, raising selection issues that affect whether the sample is representative of the target population.
Response: Thank you for this suggestion. As mentioned above, this study specifically evaluated a population of individuals with OUD because that is the population served by the primary investigator. Patients were tested for these gene variants as a part of their routine care based upon clinical symptoms observed repeatedly by the provider (e.g., low energy, low motivation) and the known prevalence of comorbid conditions associated with MTHFR mutations (e.g., depression, bipolar disorder) with OUD. The intention of these measurements as a part of routine care was to supplement with methylated folate or SAM-e if necessary. Based on the above response, we have significantly edited paragraph 4.1 (lines 226-245).
- Are you able to say anything regarding the relationship between OUD severity, profile and prevalence of specific types of MTHFR polymorphism? If not, please explain why you do not include data on clinical profile of OUD patients for the clinic when you have access to medical records.
Response: Thank you for this suggestion and question. This study was intended to be an exploratory study assessing the general prevalence of the MTHFR polymorphism in an OUD population and whether additional research may be warranted to explore this genetic association further. A future study would be needed to fully evaluate the association between MTHFR polymorphism, OUD severity, and other clinical indicators. Based upon the design of the current retrospective study, this type of evaluation was outside the scope of our objective and was not feasible based upon the number of individuals included in the study.
- The discussion of clinical implications of your findings for conditions unrelated to OUD e.g., autism, cancer is misplaced and should be removed. At the very least the connection should be anticipated in study objectives and clearly tied to your findings.
Response: We are grateful for this valuable suggestion. We agree with the reviewers. In the light of reviewer #1, we made changes in the discussion, especially the above-mentioned discussion. Thank you for this comment that strengthened our manuscript.
Round 2
Reviewer 3 Report
Reviewer is satisfied with author responses